# Effects of Yam (*Dioscorea rotundata*) Mucilage on the Physical, Rheological and Stability Characteristics of Ice Cream

**DOI:** 10.3390/polym14153142

**Published:** 2022-08-02

**Authors:** Ermides Lozano, Karen Padilla, Jairo Salcedo, Alvaro Arrieta, Ricardo Andrade-Pizarro

**Affiliations:** 1Master in Agro-Food Sciences, Department of Food Engineering, Faculty of Engineering, Universidad de Córdoba, Montería 230002, Colombia; ermideslozano@unisucre.edu.co; 2Department of Agro-Industrial Engineering, Faculty of Engineering, University of Sucre, Sucre 700003, Colombia; karenpadilla@unisucre.edu.co (K.P.); jairosalcedo@unisucre.edu.co (J.S.); alvaroarrieta@unisucre.edu.co (A.A.); 3Department of Food Engineering, Faculty of Engineering, University of Córdoba, Montería 230002, Colombia

**Keywords:** mucilage, physical stability, pseudoplastic fluid, stabilizer

## Abstract

In the present investigation, yam mucilage was evaluated as a stabilizer and emulsifier in the formulation of vanilla flavored ice cream; physicochemical, rheological, and stability characteristics were determined. A completely randomized bifactorial design was used (yam mucilage: Carboxymethylcellulose ratio with the following levels: 100:0, 80:20, 50:50, and 20:80, and stabilizers concentration with levels of 0.4 and 0.8%). Results showed an increase in the protein content present in ice cream mixture as the amount of mucilage increases. Rheologically, it was found that ice cream has the characteristic behavior of a pseudoplastic fluid, presenting a viscoelastic structure where elastic behavior predominates. In addition, ratios with a higher content of mucilage incorporated a greater volume of air and presented the longest melting times, delaying drops falling time; in the same way mucilage gives ice cream a freezing temperature between −6.1 to −2.8 °C, indicating that the application of mucilage in food industry is possible due to its nutritional value, and it gives ice cream stability properties.

## 1. Introduction

Ice cream is a liquid mixture that after agitation and simultaneous cooling is transformed into a paste [1], it is also a complex system consisting of a frozen matrix containing air bubbles, fat globules, ice crystals and a non-frozen serum phase [2]. It consists of elements of different nature such as sugars, fats, and proteins that give it the structure of colloidal dispersion and allow it to be considered a nutritious food that represents a source of proteins, some vitamins, minerals, and some phytochemicals [3,4].

During the last two decades, the science and technology of ice cream has experienced a remarkable progress in terms of the exploration and understanding of factors such as structure, texture and stability during storage, which has led to the realization of the incorporation of new additives in formulations, not only because they provide modifications for particular requirements such as viscosity increase, emulsification, and water retention, but also to improve health-related aspects [3,5,6]. Among the wide variety of additives on the market that have been exploited as thickeners, stabilizers, and emulsifying agents in food systems such as ice creams are natural gums (gum arabic, tragacanth and guar), modified gums (carboxymethylcellulose, CMC), methylcellulose and microcrystalline cellulose), some extracts (pectins), as well as native and modified starches are used, highlighting CMC as the most used stabilizer in the food industry. One way to improve the stabilizing power of these food additives is by using the mixture of two or more hydrocolloids, in order to complement them and thus improve the stability of the product, for example using CMC and some natural gums [7].

Yam (*Dioscorea* spp.) has been identified as one of the most important food crops and is a multi-species of over 600 species with subterranean tubers or rhizomes that grow in tropical and sub-tropical regions. The bioactive components of yam include mucilage polysaccharides (non-starch polysaccharides), starch, amino acids, flavonoids, allantoin, phenolic compounds, and other constituents. Presently, mucilage polysaccharides, as one of the major components of water-extracts from yam, are attracting increasing attention for their diverse bioactivity [8,9].

Mucilage is considered a hydrocolloid or gummy substance made up of proteins that interact strongly with water and cellulosic polysaccharides, containing the same number of sugars as gums and pectins, and it can be easily dissolved in water [10,11,12,13,14]. Due to these good functional properties, it is applicable in food industry, so it can be used in different matrices as a thickener, gelling, and chelating additive to avoid phase separation thanks to its complex polysaccharide nature; in addition, its intake can reduce cholesterol levels and help with intestinal function. Therefore, mucilage is an alternative as a natural hydrocolloid, now that trends are focused on consuming healthy products, to preserve the sensory and physical-chemical quality of the food, specifically as a stabilizer and emulsifier in food matrices such as ice cream [15,16]. The addition of chia seed mucilage as a stabilizer produced an ice cream with characteristics similar to those obtained with the commercial stabilizer (guar gum) [15,17]. Thus, the objective of this research was to evaluate yam mucilage as a stabilizer and emulsifier in the formulation of vanilla flavored ice cream.

## 2. Materials and Methods

### 2.1. Raw Material and Extraction of Yam Mucilage

Yam (*Dioscorea rotundata*) from the department of Sucre (Colombia) was used. Bubbling method was used for the extraction of mucilage, done with continuous pilot-scale equipment located at Unitary Operations plant of the Universidad de Sucre, which operates in environmental conditions with a yam:water ratio of 1:8 and whose basis is flotation in the presence of air. Liquid mucilage was dried in a vacuum freeze dryer (Freezone, Labconco, Kansas City, NA, USA) for 60 h; for this purpose it was first frozen at −50 °C, and after drying it was macerated for later use [13,18].

### 2.2. Formulation and Preparation of Vanilla Flavored Ice Cream

Ice creams were prepared according to the method of BahramParvar and Goff [19] with some modifications. The ingredients were mixed in the following order (whole milk, milk powder, unsalted butter, stabilizer and emulsifier, finally sugar), homogenization in a Heidolph Ultraturrax at 9000 rpm for 4 min, pasteurization at 62 °C for 30 min, then cooling and ripening, then 0.1% of vanilla essence mixture was added, finally freezing and incorporation of air, finishing with ice cream hardening. For the formulation of the ice cream, the values allowed by the Colombian standard (NTC 1239) were taken (Table 1).

### 2.3. Physicochemical Properties of Vanilla-Flavored Ice Cream

The following physicochemical tests were carried out in triplicate: Titratable acidity expressed in lactic acid, following the procedure AOAC 942.15 [20]; grease according to the Gerber 989.04 method; the pH was measured directly with a METROHM pH-meter based on the AOAC procedure method 981.12; ashes by method 923.03; total solids according to gravimetric method 925.105; and protein by the method 991.20. Protein was determined by multiplying the total nitrogen by a factor of 6.38 since it is a dairy food added with protein mixtures.

### 2.4. Rheological Behavior of Vanilla Flavored Ice Cream

Stationary and dynamic tests: Stationary and dynamic tests were carried out on the mix in a Modular Compact Rheometer MCR 302 (Anton Paar, Graz, Austria). The equipment was controlled using RheoCompass software and data were processed using Microsoft Excel 2010.

For stationary tests, 10 mL of the ice cream mixture was taken and placed in a rheometer with cylinder attachment, varying the shear gradient in a continuous ramp from 0.1 to 100 s^−1^ in ascending and then in descending form of 100 to 0.1 s^−1^. The maximum shear gradient (100 s^−1^) was maintained for two minutes. The test was performed at a temperature of 25 °C. Experimental values of flow curves were adjusted to Oswald–de Waele (Equation (1)), Bingham (Equation (2)), and Herschel–Bulkley (Equation (3)) rheological models [21,22].
σ = *K*γ^n^(1)
σ = σ_0_ + *K* γ(2)
σ = σ_0_ + *K* γ^n^
(3)
where σ is shear stress (Pa), γ is shear gradient (s^−1^), *n* is flow behavior index, *K* is consistency coefficient (Pa·s^n^), and σ_0_ is yield stress (Pa).

For dynamic tests, the viscoelastic behavior was determined operating in oscillatory mode, using parallel-plate geometry with a diameter of 25 mm (P-PTD 200/E, Anton Para, MCR 302). A gap of 1 mm between plates was fixed. Initially, an amplitude sweep was performed at a strain between 0–100% and frequency of 1.0 Hz, to determine the linear viscoelasticity region. The viscoelastic characterization of ice cream mixture was obtained by varying the frequency between 0.01 to 100 Hz [23], and rheological measurements were carried out in duplicate. Storage modulus (G’) was also obtained, a measure of the elastic property; loss modulus (G”), a measure of the viscous property; and loss tangent (tan α). The behavior of variables mentioned above was studied at 25 °C. Results of frequency sweeps were analyzed using power model.
(4)$$G\prime =K\prime \left(\omega^{n\prime }\right)$$
(5)$$G\prime \prime =K\prime \prime \left(\omega^{n\prime \prime }\right)$$
where *K*’, *K*”, *n*’, *n*” are constants and ω is the angular frequency, Hz.

### 2.5. Physical Stability of Vanilla Flavored Ice Cream

Overrun: it was measured as described by Daw and Hartel [24], taking the weight of a given volume of ice cream mix and final ice cream. Overrun measurements were taken in triplicate using Equation (6).
(6)$$Overrun\;\left(\%\right)=\frac{Weight\;of\;mix\;-Weight\;of\;ice\;cream\;}{Weight\;of\;ice\;cream}*100$$

Melting: Based on samples of hardened ice cream in storage (approximately −15 °C), an approximate mass of 45 ± 5 *g* was taken. Each sample was placed in a 2 mm wide square openings sieve, which was suspended on a previously tared beaker. Total melting time of the ice cream was recorded and weight was taken every 5 min, controlling the temperature between 24–25 °C by means of an air conditioning system [25].

Freezing point: it was determined using a cryoscope (Funke Gerber, Berlin, Germany) according to the methodology reported by Marshall and Goff [26]. The ice cream mix was diluted with ultra-pure water at a ratio 1:2, before measuring.

### 2.6. Statistical Analysis

A completely bifactorial random design was carried out where the first factor was yam mucilage:CMC ratio (M:CMC) with the following levels (100:0 M:CMC, 80:20 M:CMC, 50:50 M:CMC, 20:80 M:CMC), and the second factor was the concentration of stabilizer (CS) with the following levels (0.4 and 0.8%) and the following dependent variables: flow behavior index, consistency coefficient, storage and loss modulus, loss tangent, melting, freezing point and overrun. Analysis of variance (ANOVA) and Tukey’s test were used to compare means. Differences were considered significant at a level of *p* < 0.05. Data analysis were completed using R software version 3.1.2.

## 3. Results and Discussion

### 3.1. Physicochemical Properties of Vanilla Flavored Ice Cream

In Table 2, results obtained are listed for physicochemical characteristics of vanilla-flavored ice cream mixes. The pH values oscillated between 6.42 and 6.49, values in agreement with the ones found in the literature, where it is reported that for ice cream mixtures pH should oscillate between 6 and 7 [27]. Anova showed that the interaction of CS and M:CMC (*p* < 0.001) significantly affected pH. At low stabilizer concentration (0.4%) pH does not change in different M: CMC ratios; however, there is a slight change with an CS of 0.8. Figures found that degradation to lactose and subsequent production of lactic acid are likely to happen.

Total solid (TS) values ranged between 32.07 and 33.57%, which comply with Colombian regulations (NTC, 1239). Similar results have been reported for ice cream with probiotic bacteria 36.2% [25], and in ice cream mix stabilized with different natural gums 36.9–37.7% [23]. The CS and M:CMC interaction (*p* = 0.0002) had a significant effect on ST. High levels of concentration (0.8%) in different ratios do not affect TS, but at a low level of CS (0.4%) it shows differences, the M:CMC ratio being 80:20, which is why they would probably present smaller crystals in ice cream, because TS has to do with the size of ice crystals, since the smaller their amount, the larger their size [28].

Titratable acidity fluctuated between 0.028 and 0.084%, similar values have been found by Kavaz and Yüksel [29] that report acidity of 0.06 and 0. 09% for soft serve ice creams, but lower than those reported for ice creams where they replace non-fat milk solids, 0.12%; in the ice cream mix stabilized with tragacanth gum, 0.13%; for ice cream stabilized with a mixture of carob, karaya and guar gums, 0.12–0.14% [28]. Differences possibly due to the variety of raw materials used in the elaboration of ice cream mixtures. ANOVA shows that the CS (*p* = 0.0087) presented a significant effect on titratable acidity. With the increase of CS there is a decrease in the ice cream mixture titratable acidity. Acidity content is of vital importance, which should not be too low, since it can cause the precipitation of casein [21], but too high contributes to obtaining an excessive viscosity in the mixture [28]. It should be noted that although low acidity values were obtained in the ice cream, there were no characteristics showing this happened in the process.

Ash values were between 0.96 and 1.12%. Results similar to those were found in ice cream prepared using modified wheat gluten, 0.92% [30]. According to Anova, the CS relationship (*p* = 0.0035) and M: CMC interaction (*p* = 0. 0074) had a significant effect on ashes. It was found that for CS values of 0.4% there is no change in the different ratios, while for EC values of 0.8% with the decrease of mucilage content in M: CMC ratio there is an increase in the amount of ash in the mixture. The treatment with a 20:80 ratio and 0.8% of stabilizer, was the one with the highest ash content. Increasing the concentration decreases free water activity by increasing ash content, which is beneficial for the ice cream as ashes have physiological importance, since they participate in taste, inhibit enzymatic catalysis and other reactions that influence food texture [15].

Fat values were between 6.77 and 7.70%, which are within the ranges allowed by Colombian regulations (NTC, 1239). Data similar to that reported for soft serve ice cream, 8% [31], but lower for coconut milk-based ice cream, 20% [32]; and soft serve ice cream with probiotics, 9.32–10.07% [25]. According to ANOVA, any of the evaluated factors significantly affected fat content. The importance of the high values of fat in ice cream is that this macromolecule gives a heavier sensation to the palate and reduces the size of ice crystals, this is because fat participates in the constitution of frozen shake forming a lattice and consolidated, which improves viscosity, decreases melting, stabilizes and promotes the incorporation and dispersion of air, imparts aroma and favors the formation of ice crystals, making consistency softer and creamier, because despite the simplicity of the ingredients, the interaction between components of milk ice cream is quite complex because it is an emulsion, a foam, and a dispersion at the same time [15,33].

Protein values were between 2.68 and 3.96%, which comply with Colombian regulations for ice cream (NTC, 1239), also agrees with values reported for different ice creams between 3 and 4% [24,25]. According to ANOVA a ratio M: CMC (*p* = 0.0007) and CS (*p* = 0.0059) showed a significant effect on the protein. At an CS of 0.4%, no difference was found for protein, but as the M:CMC ratio increased, at high concentration, protein present in the ice cream mixture increased. The high values of proteins present in the ice cream mix in addition to improving the nutritional composition, helps ice cream stability as they emulsify fat and contribute to the formation of structures, including emulsification, shake and water retention capacity. Being very important for the correct development of the desired texture in the final product. Protein solubility and their interaction with other components of the mixture (polysaccharides such as mucilage) affected the ability to emulsify and stabilize fat globules that will be dispersed in this phase, to later crystallize them during the churning–freezing and stabilize at once the built-in air [31,32,33].

### 3.2. Rheological Behavior

In Figure 1, flow curves of vanilla flavored ice cream are presented. Treatments where ratios with the smallest amount of yam mucilage (20:80 and 50:50 M:CMC) were used, independent of concentrations (0.4 and 0.8%), ascent and descent curves, despite not coinciding, did not have statistically significant effect, indicating that the mixtures for ice cream were time-independent fluids. All samples of soft serve ice cream mix showed a characteristic behavior of a pseudoplastic fluid, this behavior has been reported in mixtures for ice cream exopolysaccharide producing bacterial strains [34] and mixtures with the inclusion of basil seed gum, tragacanth gum, karaya, carob, guar gum, and CMC [35,36,37]. This behavior of the ice cream mixes has attributed to involvement of partially broken-down micellar casein at the droplet surface in the ice cream mix [27].

The Oswald–de Waele model was found to be adequate to describe the flow behavior of the vanilla flavored ice cream mixes (R^2^_aj._ > 0.994). The flow behavior index (n) ranged between 0.471 and 0.995 (Table 3), which confirms the pseudoplasticity of ice cream mixes. Similar results were found in ice cream mixtures produced by exopolysaccharide strains, 0.25 and 0.98 [34]; in ice cream stabilized with different hydrocolloids, 0.55 and 0.873 [37]; in low fat ice cream, 0.474 and 0.743 [35]; and with the inclusion of basil seed gum, watercress seed gum and quince seed gum, 0.515 and 0.711 [38].

ANOVA showed that M:CMC ratios, CS and their interaction had a significant effect on flow index (n) and consistency coefficient (*K*). The flow behavior index increases with the decrease of CS and with the content of yam mucilage in the M: CMC ratio (Table 3). This reflects that as the stabilizer concentration decreases and the amount of yam mucilage increases in the ice cream mixture, it is closer to a Newtonian fluid behavior. The consistency coefficient shows no differences in the CS of 0.4%; however, it decreases as the M:CMC ratio increases in the CS of 0. 8%. In general, the lowest value of n and the highest value of *K* were obtained when the 20:80 mucilage:CMC ratio with 0.8% was used. This result may be due to a hydrocolloids nature that causes attractive protein–fat interactions, as they support pasteurization, increase viscosity by being good thickeners, and act well in acid medium [39].

The main parameters for viscoelastically characterizing fluids are storage module (*G*’) and loss module (*G*”), which were plotted as a function of frequency (Hz) (Figure 2a,b). For ice cream mixtures at low frequencies, the elastic behavior predominates over the viscous (*G*’ ≥ *G*”), so that the weak gel behavior prevails except for the mixtures with 100:0 M:CMC ratio with 0.4 and 0.8% CS, they tend to be a structured liquid, but with high frequencies all treatments tend to form a strong gel (*G*’ < *G*”). Similar results have been found in the effect of basil seed, cress seed and quince seed gums on ice cream rheological properties, where *G*’ > *G*’’, predominantly the weak gel behavior [38] and in mixtures for ice cream stabilized with natural gums individually or their combination [36,40].

Frequency sweeps results were analyzed using the power law model that showed an R^2^ ranging between 75 and 92% (Equations (4) and (5)). The estimated values of parameters are summarized in Table 4, as well as the general average of loss tangent.

The values of *K*’ and *K*” ranged between 0.755–2.201 Pa·s^n’^ and 0.131–0.907 Pa·s^n”^, respectively, and the values of n’ and n” ranged from 2.062–2.264 to 1746–2201, respectively (Table 4). The value of parameter K’ shows that the magnitudes of G’ were slightly higher than those of *G*”, rheological behavior of all ice cream mixtures is described as weak gels, also comparing values of the exponents estimated from the power law, we obtain that n’ > n”, inferring that the elastic component is more dependent on frequency than the viscous component [23,38]. In treatments that used only mucilage (100:0 ratio) in both concentrations (0.4 and 0.8%), tan (δ) and *G*’ ≤ *G*” were found to increase behavior similar to that of ice cream added with replacement of fat based on carbohydrates and proteins [41].

### 3.3. Physical Stability of Ice Cream 

Results obtained for physical stability (overrun, freezing point and melting time) of vanilla flavored ice cream are shown in Table 5. Overrun values were between 56.27 and 78.27%. Similar results have been found in the production of ice cream with chia mucilage as an emulsifying and stabilizing agent with values of 60% [15], and in ice creams with honey, trehalose and erythritol, where air incorporation was 51% [42], and higher than other stabilized ice creams with different natural gums, 29.69–33.37% [34], and 48% [25]. These differences may be due to the amount of solids present in the ice cream mix. ANOVA shows that stabilizer concentration (*p* = 0.0267) and M:CMC ratio (*p* < 0.001) significantly affected the overrun variable. When CS (0.4 to 0.8%) and M:CMC ratio increase, an increase in the overrun or amount of air incorporated occurs. The incorporation of air in the ice cream is an important physical characteristic that affects its quality, interfering in texture, smoothness and stability, and this is introduced by the shake and is a necessary ingredient, because without it, it would be too dense, hard, and cold. Furthermore, the incorporation of air depends on the composition of the mixture (fat content), as well as on the type and amount of stabilizer and emulsifier used, so that yam mucilage gives the ice cream stability properties, and this is due to protein–fat interactions, since in these interactions hydrocolloids with a high protein content such as mucilage are key to the incorporation of air and its control, because proteins, due to their emulsifying properties during the shake, contribute to the formation of air bubbles [2].

Freezing point values were between −6.1 and −2.8 °C. Similar results are reported for coconut milk-based ice cream with inulin and locust bean gum addition, −3.0 and −2.5 °C [32]. All factors and their interaction had a significant effect on freezing point. High amounts of mucilage in the M:CMC ratio (80:20 and 100:0) and an increase in CS do not produce a change in the ice cream freezing point, but at low amounts of mucilage in ratios (20:80 and 50:50) with an increase in CS produces an increase in the freezing point, possibly due to the increase of dry matter in ice cream due to the addition of mucilage, since due to its high molecular weight it contributes to the increase of dry matter. In addition, the freezing point, being a colligative property dependent on the type and content of constituents of the mixture and influenced by the number of molecules in solution, specifically solids. Trgo et al. [43] and Arellano et al. [44], suggested that in mixtures with high solids content, these variable decreases and can be lower than −3 °C, whereas in mixtures rich in fat, and low in solids it can be greater than −1.4 °C.

Melting time values were between 33.33 and 70 min. According to ANOVA, M:CMC ratio (*p* < 0.001) and interaction CS and M: CMC ratio (*p* = 0143) presented a significant effect on the melting time variable. At a high level of concentration (0.8%) the increase in the amount of mucilage in M:CMC ratio produces an increase of more than 100% in melting time; however, at a low level of concentration (0.4%) this change is not that noticeable. Mucilage has an emulsifying capacity, which gives it fat stability and retention of air bubbles, factors that decrease the melting rate due to fat and air networks within the matrix that offer high resistance to the already existing fluid, which has melted by the transfer of heat with the environment.

## 4. Conclusions

The content of mucilage contributes to the increase of protein and ash content, and rheological analysis showed that mixtures of soft serve ice cream with addition of mucilage had a characteristic behavior of pseudoplastic fluid, at low frequencies the elastic behavior predominates over the viscous, except mixtures with 100% mucilage for both levels of working concentration that tend to be a structured liquid. Sweep frequency analysis showed that mixtures behaved as weak gels, and that the elastic component is more frequency dependent than the viscous component.

Overrun, freezing point and melting, properties of physical stability, showed significant variation due to the interaction of factors, formulations with the highest mucilage content for both levels of concentration were those with the highest volume of air incorporated, melting and delay in the falling time of drops.

## Figures and Tables

**Figure 1 polymers-14-03142-f001:**
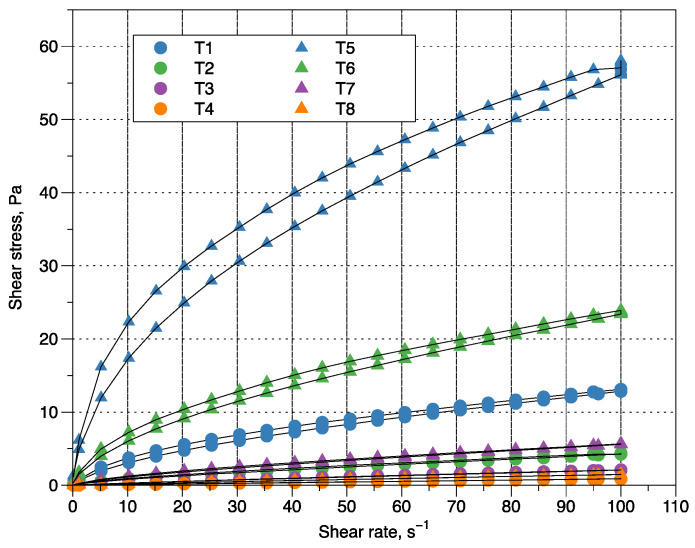
Flow behavior of vanilla flavored ice cream mixes (Concentration CS 04%, T1 (20:80 M:CMC), T2 (50:50 M:CMC), T3 (80:20 M:CMC), T4 (100:0 M:CMC)), (Concentration CS 0.8%, T5 (20:80 M:CMC), T6 (50:50 M:CMC), T7 (80:20 M:CMC) and T8 (100:0 M:CMC)).

**Figure 2 polymers-14-03142-f002:**
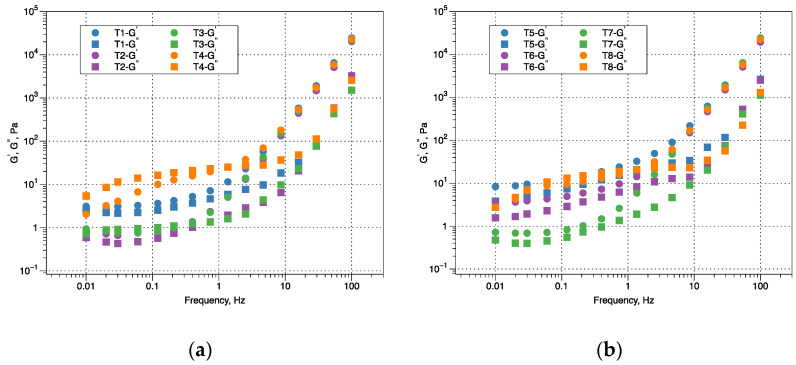
The storage (*G*’) loss (*G*”) modulus as a function of frequency of vanilla flavored ice cream mixes. (**a**) (Concentration 0.4%, T1 (20:80 mucilage:CMC), T2 (50:50 mucilage:CMC), T3 (80:20 mucilage:CMC), and T4 (100: 0 mucilage:CMC)), (**b**) (Concentration 0.8%, T5 (20:80 mucilage:CMC), T6 (50:50 mucilage:CMC), T7 (80:20 mucilage:CMC), and T8 (100:0 mucilage:CMC)).

**Table 1 polymers-14-03142-t001:** Formulation for soft serve ice cream.

Ingredients	Level 1		Level 2	
	Amount (g)	Percentage (%)	Amount (g)	Percentage (%)
Milk	1150.5	76.7	1149	76.6
Milk powder	118.5	7.9	118.5	7.9
Unsalted butter	28.5	1.9	28.5	1.9
Sugar	196.5	13.1	192	12.8
Stabilizer / emulsifier	6	0.4	12	0.8
Total	1500	100	1500	100

**Table 2 polymers-14-03142-t002:** Average values of vanilla-flavored soft serve ice cream physic-chemical characteristics.

Physicochemical Characteristics	Concentration (%)	Stabilizers Ratio (Mucilage:CMC)			
		20:80	50:50	80:20	100:0
pH	0.4	6.45 ± 0.01 ^bA^	6.44 ± 0.01 ^bA^	6.44 ± 0.02 ^aA^	6.47 ± 0.04 ^aA^
	0.8	6.49 ± 0.02 ^aA^	6.49 ± 0.02 ^aA^	6.42 ± 0.02 ^aB^	6.42 ± 0.03 ^aB^
Total solids (TS), %	0.4	32.09 ± 0.21 ^bC^	32.65 ± 0.52 ^aB^	33.57 ± 0.27 ^aA^	32.91 ± 0.18 ^aB^
	0.8	32.60 ± 0.11 ^aA^	32.39 ± 0.09 ^aA^	32.20 ± 0.04 ^bA^	32.07 ± 0.06 ^bA^
AT (% lactic acid)	0.4	0.036 ± 0.001 ^aA^	0.036 ± 0.003 ^aA^	0.084 ± 0.011 ^aA^	0.048 ± 0.011 ^aA^
	0.8	0.037 ± 0.001 ^aA^	0.035 ± 0.001 ^aA^	0.028 ± 0.008 ^bA^	0.037 ± 0.002 ^bA^
Ashes, %	0.4	1.01 ± 0.02 ^bA^	0.98 ± 0.05 ^aA^	1.04 ± 0.03 ^aA^	0.98 ± 0.07 ^aA^
	0.8	1.12 ± 0.06 ^aA^	1.05 ± 0.05 ^aAB^	0.97 ± 0.04 ^aB^	0.96 ± 0.06 ^aB^
Fat, %	0.4	7.70 ± 0.26 ^aA^	6.77 ± 0.16 ^aA^	7.47 ± 0.15 ^aA^	7.20 ± 0.26 ^aA^
	0.8	7.23 ± 0.16 ^aA^	7.03 ± 0.15 ^aA^	7.17 ± 0.21 ^aA^	7.50 ± 0.10 ^aA^
Protein, %	0.4	3.42 ± 0.20 ^aA^	3.46 ± 0.18 ^aA^	3.26 ± 0.19^aA^	3.70 ± 0.28 ^aA^
	0.8	2.68 ± 0.30 ^bB^	3.64 ± 0.11 ^aA^	3.68 ± 0.41^aA^	3.96 ± 0.34 ^aA^

Means with different lowercase letters in the same column and with different capital letters in the same row indicate a statistically significant difference according to the Tukey test (*p* ≤ 0.05).

**Table 3 polymers-14-03142-t003:** Rheological parameters estimated from Oswald–de Waele model of vanilla-flavored soft serve ice cream mixture.

Rheological Parameters	Concentration (%)	Ratio (Mucilage:CMC)			
		20:80	50:50	80:20	100: 0
n	0.4	0.586 ± 0.044 ^aD^	0.668 ± 0.031 ^aC^	0.855 ± 0.017 ^aB^	0.955 ± 0.022 ^aA^
	0.8	0.471 ± 0.046 ^bD^	0.556 ± 0.041 ^bC^	0.691 ± 0.026 ^bB^	0.9 ± 0.001 ^bA^
K (Pa·s ^n^)	0.4	0.869 ± 0.19 ^bA^	0.266 ± 0.031 ^bA^	0.04 ± 0.004 ^aA^	0.011 ± 0.001 ^aA^
	0.8	6.659 ± 1.522 ^aA^	1.859 ± 0.369 ^aB^	0.239 ± 0.029 ^aC^	0.023 ± 0.001 ^aC^

Means with different lowercase letters in the same column and with different capital letters in the same row indicate a statistically significant difference according to the Tukey test (*p* ≤ 0.05).

**Table 4 polymers-14-03142-t004:** Viscoelastic parameters estimated according to power law model, in soft serve ice cream.

Viscoelastic Parameters	Concentration (%)	Ratio (Mucilage:CMC)			
		20:80	50:50	80:20	100:0
*K*’, Pa·s^n’^	0.4	1.352 ± 0.002 ^bAB^	0.755 ± 0.001 ^bC^	1.255 ± 0.071 ^aB^	1.589 ± 0.008 ^aA^
	0.8	2.201 ± 0.046 ^aA^	1.478 ± 0.051 ^aBC^	1.234 ± 0.148 ^aC^	1.629 ± 0.123 ^aB^
*K*”, Pa·s^n^”	0.4	0.502 ± 0.078 ^aA^	0.157 ± 0.000 ^aA^	0.131 ± 0.001 ^aA^	0.369 ± 0.021 ^aA^
	0.8	0.679 ± 0.089 ^aA^	0.192 ± 0.035 ^aA^	0.347 ± 0.107 ^aA^	0.907 ± 0.084 ^aA^
N’	0.4	2.136 ± 0.014 ^aA^	2.264 ± 0.071 ^aA^	2.145 ± 0.021 ^aA^	2.130 ± 0.071 ^aA^
	0.8	2.062 ± 0.073 ^aA^	2.072 ± 0.013 ^bA^	2.155 ± 0.019 ^aA^	2.068 ± 0.017 ^aA^
N”	0.4	1.872 ± 0.047 ^aBC^	2.201 ± 0.07 ^aA^	2.031 ± 0.002 ^aAB^	1.799 ± 0.131 ^bC^
	0.8	1.792 ± 0.025 ^aB^	2.031 ± 0.014 ^bA^	1.746 ± 0.091 ^bB^	2.050 ± 0.008 ^aA^
tan (δ)	0.4	0.45 ± 0.04 ^aB^	0.42 ± 0.08 ^aB^	0.56 ± 0.07 ^aB^	1.08 ± 0.11 ^aA^
	0.8	0.39 ± 0.01 ^aB^	0.38 ± 0.01 ^aB^	0.36 ± 0.01 ^bB^	0.75 ± 0.06 ^bA^

Means with different lowercase letters in the same column and with different capital letters in the same row indicate a statistically significant difference according to the Tukey test (*p* ≤ 0.05).

**Table 5 polymers-14-03142-t005:** Physical stability variables measured for vanilla-flavored soft serve ice cream.

Variables	Concentration (CE) (%)	Ratio (M:CMC)			
		20:80	50:50	80:20	100:0
Overrum,%	0.4	60.97 ± 2.40 ^bC^	57.94 ± 2.48 ^bC^	64.95 ± 1.14 ^bB^	72.35 ± 0.98 ^bA^
	0.8	56.27 ± 1.25 ^aD^	73.76 ± 1.08 ^aB^	66.98 ± 0.83 ^aC^	78.27 ± 1.67 ^aA^
Freezing point, °C	0.4	−4.60 ± 0.46 ^aC^	−4.90 ± 0.36 ^aBA^	−5.40 ± 0.47 ^aCB^	−6.10 ± 0.55 ^aA^
	0.8	−2.80 ± 0.04 ^bB^	−3.30 ± 0.61 ^bB^	−5.70 ± 0.33 ^aA^	−6.00 ± 0.49 ^aA^
Melt time, min	0.4	56.67 ± 2.89 ^aA^	41.67 ± 2.89 ^bB^	65 ± 5 ^aA^	61.67 ± 2.5 ^bA^
	0.8	33.33 ± 5.00 ^bC^	50.00 ± 5.00 ^aAB^	53.33 ± 2.89 ^bB^	70.00 ± 5.00 ^aA^

Means with different lowercase letters in the same column and with different capital letters in the same row indicate a statistically significant difference according to the Tukey test (*p* ≤ 0.05).

## Data Availability

Not applicable.

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
