# Peer review of "Effects of Yam (Dioscorea rotundata) Mucilage on the Physical, Rheological and Stability Characteristics of Ice Cream"

_polymers, 2022, doi:10.3390/polym14153142_

Round 1
Reviewer 1 Report
Lozano et al. have investigated the quality attributes of ice cream containing yam mucilage. The manuscript is well written. I only have minor comments.
(1) A reference is required for the ice cream preparation.
(2) For the frequency sweep, the fitting might not be appropriate because there are two different zones of the curve. At low frequency region (< 1 Hz), it shows much less frequency dependence compared to that of higher frequency. This should be fitted using two separate power laws. This might need to be corrected.
Author Response
Point 1: A reference is required for the ice cream preparation.
Response 1: The reference was added (BahramParvar and Goff [19], line 80)
Point 2: For the frequency sweep, the fitting might not be appropriate because there are two different zones of the curve. At low frequency region (< 1 Hz), it shows much less frequency dependence compared to that of higher frequency. This should be fitted using two separate power laws. This might need to be corrected.
Response 2: Despite the fact that 2 zones are shown in the viscoelasticity curves (figure 2), in most of the treatments the power-law presents a coefficient of determination around 90% (90% of the variance of the dependent variable (G' and G'') is explained by the variance of the independent variable, Frequency). Due to the above and to facilitate comparison with similar investigations, we consider that a single power law should be maintained.
Reviewer 2 Report
The manuscript is about the influence of yam mucilage on the chemical and physical, properties of ice cream. The topic is interesting and of great importance, however, it is poorly written and misses experiments such as sensory evaluation which is very critical for this product. On the other hand, the results are not deeply discussed.
The manuscript is not acceptable in its present form and must be revised.
Specific Comments:
Please write about yam (botany, composition, …) and its mucilage (composition, applications, …) in the introduction.
Please add some manuscripts which have incorporated mucilage into ice cream.
Line 62: FreeZone
Table 1: what was the basis for determining the weight of ingredients? Why the weight of sugar is not equal in both formulations?
Line 77 and 79: these are chemical tests not physicochemical. Please write “chemical properties of vanilla-flavored ice cream”. Please correct it in other sections (Results and discussion, …)
Lines 126-128: paraphrase these sentences.
Line 137: please add a reference and add more detail about the experiment
Line 141: which software and tests were used for statistical analysis? Was the significance level 5%?
Please check the statistical analysis. It seems that some values do not have significant differences, but you have used different letters for them. For instance, 6.44±0.02 and 6.42±0.02 do not have significant differences, but you have used different letters for them.
Line 157: what does “EC” stand for? Please write the full name. (In some sections you have written “CE”).
Line 187: traganto gum?! Please check the name of the gum.
Line 189: “modified wheat gluten”
Line 235: exopolysaccharide producing bacterial strains
Lines 299-301: based on these rheological properties, yum mucilage cannot be a cryoprotectant (It can have negative effects and cause the formation of more ice crystals). Please check the reference and correct these sentences.
Line 316: higher or lower?
Why the freezing point has decreased from -2.8 to -6 with increasing mucilage concentration? Why the freezing point has increased with the incorporation of stabilizers?
Author Response
Point 1: Please write about yam (botany, composition, …) and its mucilage (composition, applications, …) in the introduction.
Response 1: A paragraph about yam and mucilage was added (lines 46-52)
Point 2: Please add some manuscripts which have incorporated mucilage into ice cream.
Response 2: manuscripts were added (Lines 63-65).
Point 3: Line 62: FreeZone
Response 3: change was made.
Point 4: Table 1: what was the basis for determining the weight of ingredients? Why the weight of sugar is not equal in both formulations?
Response 4: the basis for determining the weight of ingredients was the Colombian standard. The small differences in the percentages of sugar are made to maintain the amount of solids.
Point 5: Line 77 and 79: these are chemical tests not physicochemical. Please write “chemical properties of vanilla-flavored ice cream”. Please correct it in other sections (Results and discussion, …)
Response 5: Not only chemical tests were carried out on the ice cream, for which the term physicochemical properties is maintained.
Point 6: Lines 126-128: paraphrase these sentences.
Response 6: The sentences were rewritten (lines 139-141)
Point 7: Line 137: please add a reference and add more detail about the experiment
Response 7: change was made (lines 150-152)
Point 8: Line 141: which software and tests were used for statistical analysis? Was the significance level 5%?
Response 8: software and tests were added (lines 161-163). The significance level was 5% (line 162)
Point 9: Please check the statistical analysis. It seems that some values do not have significant differences, but you have used different letters for them. For instance, 6.44±0.02 and 6.42±0.02 do not have significant differences, but you have used different letters for them.
Response 9: statistical data were corrected (Table 2)
Point 10: Line 157: what does “EC” stand for? Please write the full name. (In some sections you have written “CE”).
Response 10: The terms "CE" and "EC" were replaced by "CS" (Concentration of stabilizer, line 158), this change was made in the whole text.
Point 11: Line 187: traganto gum?! Please check the name of the gum.
Response 11: traganto was replaced by tragacanth (line 194)
Point 12: Line 189: “modified wheat gluten”
Response 12: change was made (line 204)
Point 13: Line 235: exopolysaccharide producing bacterial strains
Response 13: change was made (line 250)
Point 14: Lines 299-301: based on these rheological properties, yum mucilage cannot be a cryoprotectant (It can have negative effects and cause the formation of more ice crystals). Please check the reference and correct these sentences.
Response 14: sentence was deleted
Point 15: Line 316: higher or lower?
Response 15: term was corrected (line 327)
Point 16: Why the freezing point has decreased from -2.8 to -6 with increasing mucilage concentration? Why the freezing point has increased with the incorporation of stabilizers?
Response 16: Statistical analysis showed that the interaction had a significant effect, so only this interaction was explained (lines 351-361).
Round 2
Reviewer 2 Report
The manuscript is acceptable.